# Assessing Carbohydrate Counting Accuracy: Current Limitations and Future Directions

**DOI:** 10.3390/nu16142183

**Published:** 2024-07-09

**Authors:** Débora Amorim, Francisco Miranda, Andreia Santos, Luís Graça, João Rodrigues, Mara Rocha, Maria Aurora Pereira, Clementina Sousa, Paula Felgueiras, Carlos Abreu

**Affiliations:** 1Applied Digital Transformation Laboratory (Adit-LAB), Polytechnic Institute of Viana do Castelo, Rua Escola Industrial e Comercial de Nun’Álvares, 4900-347 Viana do Castelo, Portugal; deboram@ipvc.pt; 2Polytechnic Institute of Viana do Castelo, Rua Escola Industrial e Comercial de Nun’Álvares, 4900-347 Viana do Castelo, Portugal; fmiranda@estg.ipvc.pt; 3Center for Research and Development in Mathematics and Applications (CIDMA), Department of Mathematics, University of Aveiro, 3810-193 Aveiro, Portugal; 4proMetheus, Polytechnic Institute of Viana do Castelo, Rua Escola Industrial e Comercial de Nun’Álvares, 4900-347 Viana do Castelo, Portugal; 5School of Health of the Polytechnic Institute of Viana do Castelo, Rua Escola Industrial e Comercial de Nun’Alvares, 4900-347 Viana do Castelo, Portugal; andysantos66@gmail.com (A.S.); paula.felgueiras@ulsam.min-saude.pt (P.F.); 6Health Sciences Research Unit: Nursing (UICISA: E), School of Health of the Polytechnic Institute of Viana do Castelo, Rua Escola Industrial e Comercial de Nun’Alvares, 4900-347 Viana do Castelo, Portugal; luisgraca@ess.ipvc.pt (L.G.); mararocha@ess.ipvc.pt (M.R.); aurorapereira@ess.ipvc.pt (M.A.P.); clementinalongarito@ess.ipvc.pt (C.S.); 7Center for Translational Health and Medical Biotechnology Research (TBIO)/Health Research Network (RISE-Health), School of Health of the Polytechnic Institute of Porto, Rua Dr. António Bernardino de Almeida 400, 4200-072 Porto, Portugal; joao.r.s@ess.ipvc.pt; 8Center for MicroElectroMechanical Systems (CMEMS-UMINHO), University of Minho, Campus Azurém, 4800-058 Guimarães, Portugal

**Keywords:** carbohydrate counting, diabetes mellitus, insulin therapy, personalized medicine, nutritional education

## Abstract

Diabetes mellitus is a prevalent chronic autoimmune disease with a high impact on global health, affecting millions of adults and resulting in significant morbidity and mortality. Achieving optimal blood glucose levels is crucial for diabetes management to prevent acute and long-term complications. Carbohydrate counting (CC) is widely used by patients with type 1 diabetes to adjust prandial insulin bolus doses based on estimated carbohydrate content, contributing to better glycemic control and improved quality of life. However, accurately estimating the carbohydrate content of meals remains challenging for patients, leading to errors in bolus insulin dosing. This review explores the current limitations and challenges in CC accuracy and emphasizes the importance of personalized educational programs to enhance patients’ abilities in carbohydrate estimation. Existing tools for assessing patient learning outcomes in CC are discussed, highlighting the need for individualized approaches tailored to each patient’s needs. A comprehensive review of the relevant literature was conducted to identify educational programs and assessment tools dedicated to training diabetes patients on carbohydrate counting. The research aims to provide insights into the benefits and limitations of existing tools and identifies future research directions to advance personalized CC training approaches. By adopting a personalized approach to CC education and assessment, healthcare professionals can empower patients to achieve better glycemic control and improve diabetes management. Moreover, this review identifies potential avenues for future research, paving the way for advancements in personalized CC training and assessment approaches and further enhancing diabetes management strategies.

## 1. Introduction

Diabetes mellitus (DM) is a chronic autoimmune disease affecting approximately 537 million adults, and it was responsible for 6.7 million deaths worldwide in 2021. Moreover, estimates point out that diabetes prevalence will rise to 783 million adults by 2045 [1]. These numbers highlight the relevance of diabetes in our society. In particular, a need not only for pharmacological treatments but also for prevention and new approaches to help patients manage diabetes, in favor of better metabolic control, prevention of complications, and improved quality of life. The cornerstone for patients having diabetes is to achieve on-target blood glucose (BG) levels to avoid hypo- or hyperglycemia events, which can lead to acute and long-term complications, respectively. Hypoglycemia, a condition where BG is below normal (<70 mg/dL), can result in immediate complications such as palpitations, tremors, hunger, or sweating. Moreover, severe hypoglycemia events may lead to behavioral changes, visual disturbances, seizures, loss of consciousness, coma, or even death. In the most severe cases of hypoglycemia, it is imperative to treat it immediately. For this reason, patients usually fear hypoglycemia events the most [2,3]. Generally, hypoglycemia in diabetic patients happens when there is a mismatch between the amount of insulin or hypoglycemic medication taken and the body’s actual need for it [4]. On the other hand, hyperglycemia occurs when the BG is above normal (>180 mg/dL). Hyperglycemia symptoms include intense thirst, uncontrolled weight loss, or frequent urination. However, unlike hypoglycemia, life-threatening complications associated with hyperglycemia events are not imminent. Hyperglycemia is associated with long-term damage or failure of several organs, especially the eyes, kidneys, nerves, and the cardiovascular system [5,6]. Indeed, micro- and macrovascular complications are the most common underlying cause of death for people having diabetes [7,8]. Therefore, managing diabetes to achieve on-target BG levels is paramount.

Regular physical activity plays a significant role in DM management, contributing to better glycemic control. Furthermore, meals seem to be responsible for the most significant changes in BG levels [9]. Consequently, T1DM patients combine intensive insulin regimens with proper diet management to achieve on-target BG levels. For this, patients use carbohydrate counting (CC) to dose the prandial insulin bolus, and consequently, control postprandial BG levels. CC is widely used to adjust the prandial insulin bolus to the estimated digestible carbohydrate (CHO) content of each meal. It relies on the fact that CHOs are considered the most influencing macronutrient on postprandial glucose excursions [10,11].

CC has the potential to help patients control their BG and achieve a better quality of life. However, the bolus insulin dose must be accurate to obtain maximum benefits from CC. Unfortunately, accurately determining the bolus is difficult. It depends on several factors, the insulin-to-carbohydrate ratio (ICR), the insulin sensitivity factor (ISF), the preprandial blood glucose level, the insulin remaining active from the last bolus, and mostly on the patient’s ability to correctly estimate the CHO content of each meal [12,13,14,15]. Unfortunately, accurately estimating the CHO content of each meal is challenging for the patient.

In addition to the relevance of accurately estimating the CHO content of meals to improve glycemic control, estimation errors are frequent among patients and caregivers. Indeed, it is a complex task, demanding adherence and education [13]. Therefore, CC education provided by healthcare professionals is crucial and needs to be continuous. In the literature there are plenty of educational programs targeting DM patients, some focusing on teaching how to use CC [16,17]. Regardless of the educational program, several clinical trials show improvements in glycemic control of patients who received training in CC [18,19,20,21].

A fundamental aspect of such educational programs is the tool used to assess the learning outcomes of patients on training. Existing tools, such as the AdultCarbQuiz, were designed to be patient-independent. Therefore, they are unable to fit the assessment criteria to the particular needs of each patient. Personalized educational programs and assessment tools suited to train and evaluate patients’ ability to estimate the CHO content of each meal based on their particular needs could be of great value [22]. CC is a tool that allows for variability and flexibility in dietary choices. This work reviews existing educational programs and assessment tools dedicated to training DM patients on CC. The aim of our research is to obtain a clear understanding of the benefits and limitations of existing tools and point to new research directions in view of a personalized approach.

This research was conducted following the Cochrane Collaboration and the PRISMA statements, using Rayyan [23,24]. Relevant studies were identified by searching the digital databases in PubMed, Science Direct, Scopus, and IEEE Xplore. These databases were inspected using the following query “(“Carbohydrate Counting” OR “Carbohydrate Estimates”) AND (Diabetes OR Accuracy OR Education OR Modeling)”, from 2010 until 2021, obtaining a total of 933 works after excluding duplicates. Irrelevant studies were eliminated after reading the titles and abstracts, and 33 works were included in this study after reading of the full text and being marked as “included” by all collaborators.

The remainder of this paper is structured in five sections. Section 2 describes the role of CHO intake in BG excursions and how counting carbohydrates can be relevant to achieving better glycemic control. The Section 3 highlights the relevance of health and nutritional literacy to help patients control their BG and describes existing educational programs. The tools used to assess the learning outcomes in CC are outlined in the Section 4. Finally, we propose some future research directions Section 5 and conclusions in the Section 6.

## 2. The Role of Carbohydrate Counting on Glycemic Control in Patients with T1DM

Intensive insulin therapy relies on the correct administration of insulin according to the daily needs of each patient, including basal and prandial insulin doses. Therefore, carbohydrate counting is an essential tool to help patients fit the prandial bolus. Due to its relevance, it is included in therapeutic guidelines by several international entities such as The American Diabetes Association (ADA) and Diabetes UK [25,26,27].

Carbohydrate counting is a fundamental nutritional tool, especially for T1DM patients on intensive insulin therapy. Indeed, the Diabetes Control and Complications Trial found a correlation between intensive insulin therapy and better glycemic control, expressed by lower glycated hemoglobin (HbA1c) levels [28,29]. ADA guidelines recommend an HbA1c1 level under 7.0% to keep proper glycemic control and improve the patient’s quality of life. On the contrary, a higher probability of developing microvascular complications correlates with higher values of HbA1c [25]. However, to succeed, patients on intensive insulin therapy must accurately estimate the carbohydrate content of each meal [30].

Among all the nutrients, CHOs are the most determining for postprandial glycemia. Most of the CHOs ingested are transformed into glucose in a period of time between 15 min and 2 h. Thus, carbohydrate counting becomes crucial to managing diabetes [31].

There are two levels of carbohydrate counting, the basic level, and the advanced level. As its name suggests, Basic Carbohydrate Counting (BCC) is simpler to use but more error-prone. On the other hand, Advanced Carbohydrate Counting (ACC) is more accurate but complex, requiring great dedication. Both methods should be advised, depending on the type of diabetes and on the intrinsic characteristics of each patient, e.g., personal routine and lifestyle [6,32].

Basic carbohydrate counting, known as the exchange system, is suitable for T2DM patients who are not on intensive insulin regimens. At this level, patients must ingest a consistent amount of carbohydrates per meal, following a well-established daily routine [31]. Food is organized according to its carbohydrate content and classified on a table, forming an exchange list. Depending on the patient’s context (e.g., their socio-cultural habits, the effects of medication, or the medical guidelines adopted by the healthcare team), each serving equals 10 g or 15 g of CHOs, which is considered an exchange unit. To correctly use the exchange system, patients must identify the food containing carbohydrates, read nutritional labels, understand the concept of carbohydrate exchanges, and control portion sizes. Controlling portion sizes is very important to manage the amount of food in each meal. For example, a portion size containing 45 g of CHOs may use three servings of different foods from the exchange list. In this way, patients can vary between foods with the same amount of CHOs.

On the other hand, ACC is a systematic method that requires adjusting the prandial insulin dose to the CHO intake per meal. ACC is recommended for managing T1DM, where patients are treated with continuous subcutaneous insulin infusion (CSII) or multiple daily injections (MDIs). Regardless of the treatment method, T1DM requires a basal–bolus regimen, where slow-acting insulin regulates the blood glucose levels between meals, and fast-acting insulin (bolus insulin) to keep the postprandial BG on target, according to the CHO intake of each meal [15].

The initial steps to perform ACC are the same as those of the BCC method, identify food containing CHOs, read food labels, and practice portion control. The difference is the need for monitoring pre- and postprandial BG levels and knowing the ISF and ICR values to be able to calculate the prandial bolus of insulin [32]. It is important to note that several factors such as physical activity and health status may affect the ISF and ICR. Therefore, the patient must be aware of this and adjust the ICR and ISF values if needed [33]. The calculation of bolus insulin should follow Formulas (Equation 1) and (Equation 2) [34]:(1)Bolus=Mealinsulin+Correctioninsulin
(2)$$\mathrm{Bolus}=\frac{\mathrm{CHO}}{\mathrm{ICR}}+\frac{{\mathrm{BG}}_{\mathrm{preprandial}}-\mathrm{BGT}}{\mathrm{ISF}},$$
where ${\mathrm{BG}}_{\mathrm{preprandial}}$ is the BG value measured before the meal and BGT is the blood glucose target.

Once the patient achieves proficiency in CC, it allows more flexibility in food choices and meal schedules. Moreover, CC is associated with better glycemic control, expressed by lower levels of HbA1c, and improvements in patients’ quality of life, as pointed out by the Diabetes Control and Complications Trial (DCCT) [35]. Since the DCCT, several other studies evaluated the impact of using CC to improve glycemic control in patients within different age groups.

### 2.1. Carbohydrate Counting and Glycemic Control in Children and Adolescents

Gökşen et al., in [36], investigated the effects of CC on metabolic control, body measurements, and serum lipid levels in children and adolescents with T1DM aged between 7 and 18 years. The children and adolescents were divided into two groups: a CC group and a control group. After following the groups for two years, they found HbA1c levels to be significantly lower in the CC group while no increase in weight or insulin requirements were observed. In addition to assessing metabolic control, Donzeau and colleagues [18] investigated the correlation between using CC instead of standard nutrition in the quality of life of children. They also concluded that CC could be associated with better metabolic control and high quality-of-life scores.

Rabbone et al. followed a different approach. They investigated the impact of combining CC with an automated bolus calculator (ABC) to help children treated with MDI control their BG levels. After an 18-month observational study, they highlighted that the use of an ABC led to a decrease in glycemic variability [37].

Another study, performed by Albuquerque et al., included 28 adolescents aged between 10 and 19 years treated with short- and intermediate-acting insulins on a four-month randomized clinical trial. The authors found that CC combined with fast-acting and intermediate-acting insulins favored glycemic control, decreased caloric intake without changing body composition, and allowed flexibility in CHO intake [19].

### 2.2. Carbohydrate Counting and Glycemic Control in Adults

Over the last years, several research groups investigated the effects of CC on adults. Most of them tried to understand how CC relates to improved glycemic control and better quality of life. On this line of reasoning, the GIOCAR trial [10] followed 61 adults with T1DM treated with continuous subcutaneous insulin infusion (CSII). After 24 weeks, the results showed that CC is safe, reduces the HbA1c, and improves the patients’ quality of life. In a smaller trial, Shiraishi et al. [38] found similar results when studying seven adults with T1DM. Further investigations and systematic reviews also strengthen the results showing a positive impact of CC in glycemic control and patients’ quality of life [20,33,39]. According to Pearson et al. [40], more than one-third of hospitalized patients with diabetes have hyperglycemia. Therefore, aiming to improve the glycemic control of such patients, a pilot study was designed to investigate the effects of postmeal insulin dosing based on CC instead of the established process of scheduled premeal insulin dosing without taking into account the CHO ingested by the patients. The results revealed improved glycemic control among general surgery patients. However, the same did not happen among cardiovascular surgery patients. Most patients on cardiovascular surgery had T2DM and insulin resistance, which the authors considered an explanation for this finding. In addition, the authors report an increase in nursing satisfaction due to the improved BG levels, without any occurrence of hypoglycemic events, using fewer insulin doses on the treatments.

Souto et al., in [41], studied a group of 33 T1DM patients to compare the impact of BCC and ACC regarding anthropometric, biochemical, and dietary variables. Contrary to most of the literature, the authors reported that, when compared with BCC, ACC did not improve the glycemic and lipid control of patients. Moreover, in the same context, the authors noted that ACC could increase food intake, and therefore, the body mass index and the waist circumference. However, the authors pointed out limitations to their study that might justify the contradictory results. They used a small sample of non-random patients already using CC before the trial, chosen for convenience. Additionally, they used a non-validated questionnaire to assess the patients and have identified some issues related to patients’ ability to estimate portion sizes. These methodological considerations, including sample size, the non-random nature of participant selection, and the use of non-validated instruments, underscore the need for caution in generalizing the study’s outcomes.

Aiming to obtain consistent evidence regarding the effectiveness of CC on the glycemic control of T1DM patients, Fu et al. [42] conducted a systematic review and meta-analysis comprising ten studies published between 2000 and 2014. These studies involved 773 participants of all ages. The results pointed out that, in general, compared with other diet planning methods, CC significantly contributes to reducing HbA1c. More precisely, CC significantly reduces HbA1c in adults while not in adolescents and children. The authors suggest that this could result from the adults’ improved skills to learn and apply CC compared to adolescents and children.

In a more recent systematic review and meta-analysis, performed in 2018, Vaz et al. [35] assessed the effectiveness and safety of CC in the treatment of T1DM adult patients. The authors included five randomized studies comparing CC with general dietary advice and concluded that evidence shows that CC could be beneficial to treating T1DM in adults. However, its benefit is limited to reducing HbA1c. As demonstrated in the previous discussion, CC is an essential tool to help DM patients achieve on-target BG levels. However, despite the evident benefit of using this meal planning tool, many persons consider it difficult and frequently commit estimation errors that may result in hyper- or hypoglycemic events.

## 3. Nutritional Education and Carbohydrate Counting Tools

Nutrition is relevant not only for preventing and delaying chronic diseases, but it has also been demonstrated to be a cost-effective adjunctive treatment. There is a consensus regarding the positive impact of utilizing carbohydrate counting as a nutritional tool in diabetes management. However, due to its complexity, educational programs are essential to enhance nutritional literacy and guide patients toward healthier decisions, thereby improving health outcomes and their quality of life. Enhancing diabetes management literacy stands as a crucial objective of the healthcare team, with the goal of optimizing health status and quality of life.

The task of educating patients in diabetes management requires a multidisciplinary healthcare team capable of motivating and communicating effectively with individuals from different backgrounds. It is important that all members of the healthcare team follow an established methodology to educate patients and caregivers in CC. Intervention should be continuous and person-centered, taking into account their preferences, needs, and values. It should also be adapted to their culture, context, professional activity, physical activity habits, and associated health conditions The main objective is to provide the energy and nutritional intake necessary for glycemic control and healthy eating patterns [43]. For instance, exchange lists must be designed for a specific population according to their dietary habits and culture. If adequate material is not provided, patients may obtain information from different sources that may be contradictory and make the process harder [31]. Healthcare teams should avoid the tendency to generalize nutritional plans for diabetics. Studies show that orientations must be personalized and often reassessed. Individual plans should consider stage and type of diabetes, additional health issues, food preferences, and socioeconomic background. The reassessment must be made along with the patient, especially during times of changing health status and life stages, allowing them to achieve health goals [44]. In the prescription of the dietary plan, it is important to take into account the energy supply, nutrient diversity, and the maintenance of mental, physical, and social health. The aim is to establish a complete, balanced, and varied diet [43]. There are two types of carbohydrate counting (CC), namely, basic counting and advanced counting. Nutritional education in basic counting aims to identify foods that contain carbohydrates and provide health education on fixed carbohydrate amounts for each meal. On the other hand, advanced counting involves quantifying the carbohydrate content in each meal to adjust the insulin dose, facilitating meal management, food choices, and quantities. In health education, the healthcare team should make use of tools such as lists of foods with known quantities, weighing scales, and standardized measurements. Counting in grams, compared to food exchange lists, provides more precise information but is more demanding. With developed expertise, weighing can be dispensed with as long as there is a good understanding of food quantification. Alternatively, the use of standard measures and household measures can also provide approximate values of carbohydrates. For processed and packaged foods, it is crucial to have literacy about the quantity of carbohydrates per 100 g or milliliters of food [31]. In order to promote better management of diabetes mellitus, technology has been developed to provide access to applications that facilitate carbohydrate counting and help reduce counting errors.

### 3.1. The Importance of Patients’ Nutritional Literacy

The World Health Organization (WHO) has defined health literacy as “the personal knowledge and competencies that accumulate through daily activities, social interactions and across generations. Personal knowledge and competencies are mediated by the organizational structures and availability of resources that enable people to access, understand, appraise and use information and services in ways that promote and maintain good health and well-being for themselves and those around them” [45]. Thus, poor health literacy affects the healthcare experience at all levels, individuals, providers, and the healthcare environment. Moreover, it impacts the communication between the provider and the patient, affecting their ability to access and use the healthcare system [46].

Nutrition and food literacy—specific forms of health literacy—have become increasingly relevant concepts in health promotion. Nutrition literacy could be defined as the individual’s ability to obtain, process, and understand nutritional information and to adopt appropriate and healthy eating habits [47,48,49]. Definitions of food literacy incorporate a broader spectrum of theoretical and practical knowledge and skills, including applying information on food choices and critically reflecting on the effect of food choices on personal health and society [48].

Diabetes is a chronic condition for which nutrition is crucial for prevention and management [48,50]. Indeed, diabetes prevention and management rely on three essential pillars: food, physical activity, and treatment. Consequently, proper diabetes management is a complex task requiring health literacy skills related to numeracy, reading, and comprehension [51]. People with diabetes need education and training to understand the nutritional composition of the foods they consume, and thus, make healthier food choices. To that end, patients and healthcare providers must work together to improve patients’ nutritional literacy. In particular, addressing subjects such as understanding food labels, food composition, food selection, and portion dimensioning based on nutritional information is fundamental to daily meal planning. Patients must also be aware of the advantages of eating healthy foods and the drawbacks of processed foods, alcohol, and soft beverages.

Recent studies demonstrate that health and nutritional literacy play a relevant role in the knowledge and control of diabetes. Indeed, higher levels of health literacy relate to better diabetes knowledge, and patients with a lower level of health literacy reveal more difficulties in understanding nutrition-related issues [49,51,52].

The correlation between health literacy and glycemic control is substantiated by recent studies examining adults with type 2 diabetes mellitus (T2DM). In one study involving 280 T2DM respondents, inadequate health literacy was prevalent, particularly among females, housewives, those with lower education, recipients of oral antidiabetic therapy, and those with shorter diabetes duration. Respondents with inadequate health literacy were significantly older and had higher HbA1c levels than those with marginal and adequate health literacy [52].

In another study, involving 381 participants from diabetes clinics, nearly half of the participants had inadequate health literacy, while 40% exhibited sufficient health literacy [53]. The most significant predictors of health literacy were educational level, household monthly net income, and family history of diabetes. Medication-taking emerged as the strongest aspect of self-management, with only 27% of participants achieving good glycemic control. Health literacy demonstrated a significant positive correlation with diabetes self-management and a negative correlation with HbA1c levels. The findings of this study underscore the importance of health literacy in encouraging self-management behaviors and maintaining optimal diabetes control. The recommendation that healthcare professionals assess health literacy in individuals with diabetes highlights the practical significance of these findings in informing personalized interventions aimed at improving health literacy and ultimately enhancing glycemic control outcomes. These findings emphasize the crucial role of assessing and addressing health literacy challenges for enhancing self-management and optimizing glycemic outcomes. However, a more comprehensive understanding of health literacy’s role in nutrition and the intricate relationship between interventions and health outcomes necessitates further research, as highlighted in the existing literature [46,51].

Despite its benefits, promoting nutritional literacy is not an easy task. It involves several players in different stages. It is necessary to identify population groups with low nutritional literacy, implement specific nutrition education programs and actions, and engage participants. Additionally, developing adequate and motivating educational and training material is paramount. Fortunately, recent information and communication technologies provide new ways and tools to improve patients’ adherence to educational programs [49].

The Diabetes Literacy and Numeracy Education Toolkit (DLNET) aims to fulfill such requirements. It seeks to help people with low health literacy and numeracy manage their diabetes. The DLNET contains 24 interactive modules in which the content is limited to the essential, written in plain language with short phrases, and targeted at a reading level of 4th to 6th grade. Further, color-coding and illustrations help to improve adherence and engage patients [54]. The Partnership to Improve Diabetes Education (PRIDE) Toolkit recently updated DLNET to include a Spanish version and new health behavior change targets [55]. Actually, the authors did more than translate the text; they crafted a new version that took into account the cultural differences and focused on audience needs. To objectively assess the PRIDE’s suitability to be used by different patient groups, the Suitability Assessment of Materials (SAM) instrument was used. On the evaluation, all the 30 modules of PRIDE received a classification of “superior”, i.e., SAM > 70%.

In addition to patients’ education, several tools and methods are available to help patients with diabetes management. In particular, regarding blood glucose monitoring, carbohydrate counting, and insulin dosing. The following section provides a better understanding of the tools available to help patients improve accuracy when carbohydrate counting.

### 3.2. Tools and Methods Used to Assist Patients in Carbohydrate Counting

From the previous discussion, it is clear that patients’ education is crucial for proper diabetes management. Regardless of the method used to manage meals and control CHO intake, it is well known that accuracy can have a significant impact on the metabolic control and quality of life of patients with diabetes. Existing evidence supports carbohydrate counting as the most effective meal-planning tool to manage CHO intake and reduce HbA1c concentration [14,42]. Such positive results could be because nutritional education is included in strategies to increase CC skills [56]. However, carbohydrate counting is far from an easy task. Therefore, the need for new tools to help patients achieve more accurate results is on the table.

Since carbohydrate counting is a complex task, some instruments can help patients. Nowadays, one of the most used is the insulin pump bolus wizard [57], according to many international consensus recommendations. Nevertheless, this does not substitute the need for a rigorous education in CC once the bolus wizard requires the input of CHO meal content to calculate the prandial insulin bolus [14]. In addition the well-established metabolic benefits, when performed accordingly, CC allows more flexibility in food choice, meals schedule, and frequency [14,42,56], improving the quality of life of patients [58].

In addition to the pump bolus wizard, automatic insulin dose calculators are another valuable tool used to increase the success of clinical interventions [14]. In [59,60], the authors investigated the efficacy of a bolus calculator (BC) in T1DM patients treated with MDI and CSII. In both cases, they separated the subjects into two groups, one using a BC and a control group without a BC. At the end of the trial, the authors found that the hemoglobin A1c levels had significantly decreased in the BC groups and concluded that BC is a valuable tool to help T1DM patients to achieve optimal glycemic control. Similar to the insulin pump bolus wizard, bolus calculators also require CHO content to be introduced manually by the user. Therefore, to help patients estimate the CHO content of meals, some mobile applications (apps) intend to perform this automatically, making life easier for these patients. Indeed, some studies show that using apps such as GoCARBS, BE(AR), and iSpy reduce the CHO estimation errors [61,62,63]. In addition to the ones mentioned, Doupis et al. list a range of apps for diabetes management, including those specifically targeted towards T1DM. These include Intelligent Diabetes Management, Glucose Buddy, Diabetes Manager, Diabetes Diary, Dbees, Diabetes Interactive Diary, D-Partner, and VoiceDiab [64]. It is crucial to emphasize that users should verify whether diabetes management applications have been validated by relevant health regulatory authorities in their country for safety and reliability.

To estimate the CHO content, GoCARBS requires two complementary pictures of a meal’s plate and a reference object. Then, it uses computer vision and an extensive food database to segment, recognize, and reconstruct the different food items on the plate. A comparative study concluded that patients using GoCARB obtained an absolute error of 12.28 g, which was significantly lower than their estimation, showing an error of 27.89 g [61]. Vasiloglou et al. studied the precision of this app when compared with a group of experienced nutritionists. In this study, the mean absolute error of both was around 15 g. However, the app demonstrates difficulty estimating foods such as rice, pasta, and potatoes. Moreover, its meals database is limited, making some foods even more difficult to recognize [65]. Using the same approach, iSpy seeks to help children with T1DM by using computer vision and AI to compute the CHO content of meals, along with the possibility of using text or voice descriptions to improve its performance. iSpy was associated with better CC accuracy once it reduced the frequency of individual errors greater than 10 g [63].

In its turn, BE(AR) is an app that uses augmented reality to calculate a meal’s CHO content based on the volume of the food in the virtual environment, which allows it to find the food weight [62]. Six T1DM patients used BE(AR) for three weeks in a trial and despite the reported problems related to the app’s usability, 44% of the estimations performed reduced the CC error by at least 6 g. Apps can be helpful in the CC process. However, they do not avoid the need for patient intervention or the need for continuous education of patients and caregivers in the case of children below 12 years old. Thus, lifelong education is strongly recommended, starting as early as possible [58].

To provide a comprehensive overview and facilitate comparison of the tools discussed, Table 1 presents a concise summary highlighting their main functionalities and relevant features.

## 4. Assessing the Accuracy of Carbohydrate Counting Estimations

The complexity of CC requires nutritional literacy, numeracy skills, and experience to estimate the amount of CHO consumed. Patients often commit estimation errors due to fear of hypoglycemia, difficulty in understanding the process, and the diversity of foods and cooking methods [13]. To address these issues, educational programs aimed at improving CC skills are widely recommended and should include assessment as an integral part. Several studies have been published in the last decade to assess the accuracy of this process. The most relevant will be addressed in the following subsection.

### Tools and Methods Used in Carbohydrate Counting Assessment

Assessing CC accuracy ensures that learning goals are reached and also helps healthcare providers to understand which key points they should insist on more or if they need to adapt the teaching approach. Though, there are no standardized measures to evaluate the precision with which diabetic patients estimate CHO content. The fact that there is no way to know the real carbohydrate content of a meal is pointed out as an obstacle to the assessment. Typically, the values used to compare and consider real are estimated by a specialized team with high experience in CC.

Some questionnaires were developed and validated to assess the knowledge of CC in youth and adults with T1DM. In 2008, a tool called CarbQuiz was designed to evaluate proficiency in CC. This questionnaire was validated through its implementation in 100 adults and 15 nutritionists, of whom 90% were type 2 diabetics [72]. However, the CarbQuiz has some limitations, as it assumes that patients have similar responses to CHO intake, physical activity, and BG levels, which often is not the reality [73]. Afterward, a questionnaire focused on children with T1DM was created—the PedCarbQuiz. This tool emphasizes the capacity of recognizing food containing CHO, counting CHO, and its inclusion in the insulin dose calculation. To validate this test, they correlated the scores with HbA1c values and with experts’ assessments. As expected, high scores were associated with lower HbA1c levels, and they found a statistically significant correlation between PedCarbQuiz and experts’ scores [74]. Other versions emerged, suitable for different age groups, cultures, and populations, such as the AdultCarbQuiz intended for adults with diabetes [75] and the AusPCQ, the Australian version of the American PedCarbQuiz [76].

Many authors studied groups of patients to verify estimation errors among youth and adults with type 1 and type 2 diabetes, following different methodologies. Meade and Rushton designed and applied tests to 61 subjects with type 1 and type 2 DM treated with MDI or CSII, aiming to find how accurately patients count CHOs. They described an average test score of 44% for all patients or 59% when they excluded responses about foods that patients reported as never eaten [77]. According to Brazeau et al., adult patients with T1DM, in real-life conditions, committed estimation errors around 20% of the total CHO content per meal, which represents 15.4 ± 7.8 g. They established that 63% of the meals (448) were underestimated [12]. Regarding the impact of these mistakes on glycemic control, they verified that patients with higher errors had higher glucose fluctuations. These findings were compatible with other studies also showing that underestimation errors are more common than overestimation [78,79]. Other researchers verified that inexperienced subjects frequently misinterpret high-calorie dishes with large amounts of CHOs, which they have linked to overestimations [12,79]. Studies intended for children and adolescents, as Deed et al. in [79], show that children also underestimate more often. In this study, the authors considered correctly estimating the meals within 20% of the estimation of experienced dietitians. They observed that 67% of the total meals were accurately counted, and 63 of 165 meals were underestimated and associated with high postprandial glucose. In [78], the authors examined the issue from a different perspective. They assessed the CC accuracy in adolescents with T1DM based on the premise that errors up to 10 g of CHO per meal have no consequences and above 20 g should cause postprandial hypo- and hyperglycemia, according to the conclusions of Smart et al.’s investigations [80,81]. They applied the PedCarbQuiz to 140 adolescents and found that 86% were accurate within 20 g of a meal’s CHO content, but only 42% were within 10 g [78]. In this report, they also support that underestimation is more frequent. After reviewing all these studies, it became apparent that many patients make estimation errors that can lead to glucose excursions and poor glycemic control. To change this tendency, clinicians need a validated tool to assess individual patient knowledge and establish personalized error limits [82]. This would allow personalized goals and education programs tailored to the individual’s insulin sensitivity. For example, a patient with a good understanding of CC, who can estimate the carbohydrate content within 10–15 g error, following general guidelines, may still experience hypo- or hyperglycemic events depending on their ICR and ISF. On the other hand, those with a higher margin of error may be able to follow less restrictive guidelines, which can improve treatment adherence. Therefore, adjusting the education program to the individual is essential for the proper management of diabetes.

## 5. Discussion

The thorough research conducted provided a broad understanding of the current approaches to education in CC and the available tools to assist DM patients in this endeavor. Analyzing the patients’ accuracy in CC is also extremely important. For that reason, several authors developed works on this topic. This section will address the limitations involved in the process of CC education and the assessment of patient proficiency. Additionally, the authors will suggest future directions to overcome the identified issues, focusing on improving the skills and the patient experience when using CC.

The results are not unanimous, and there is heterogeneity in the findings. Carbohydrate counting has been found to have an effect on HbA1c control, improvement in quality of life, treatment satisfaction, and psychological well-being, while also increasing flexibility in food choices [83]. However, there are divergent studies where the impact on hypoglycemic events shows heterogeneous results [42]. CC can pose a challenge for patients, even though they recognize its importance in improving glycemic control. This can result in patients demonstrating low accuracy in applying the method.

DM is a chronic disease that is affecting an increasing number of people in the population, requiring lifelong treatment combining pharmacological measures, diet, and physical activity. This disease has significant implications for managing daily activities, treatment adherence, and quality of life. Maintaining disease control within safe values requires empowering patients in the three pillars of treatment, with nutrition playing a particularly relevant role, necessitating a patient-centered multidisciplinary intervention.

The goal of treatment is to develop health literacy, including nutritional literacy, so that patients can make rational, appropriate, and varied food choices. It is necessary to adapt the intervention to the individual, taking into account their life context, including their cultural background [31,43].

This disease requires lifestyle changes and new learning, which can often be challenging for patients. In terms of nutrition, it is essential to provide adequate energy and nutrient intake, and based on this, empower the patient to administer insulin appropriately according to carbohydrate intake [31]. Among the different methods, the most common one is basic carbohydrate counting, which has limitations in terms of standardized meal patterns and restricting patient choices. In contrast, advanced carbohydrate counting allows for greater variety in choices and better metabolic control, including HbA1c. In the initial stages, the goal is to empower the patient to perform accurate counting, which may lead to decreased adherence. However, in the long term, the aim is to achieve intuitive carbohydrate control and identification. CC, compared to other methods, has shown to be more effective in reducing HbA1c, and with the development of technologies, there is a growing interest in using apps for diabetes management.

### 5.1. Current Limitations

The positive contribution of the nutritional tool carbohydrate counting, especially in T1DM, is well established. However, based on the analysis of the relevant literature, it is evident that the CC task is complex and many patients struggle to complete it successfully.

Technological advances have allowed the creation of tools to support the daily management of diabetes. Insulin pumps with bolus wizards and automatic bolus calculators are examples. Despite the benefits, insulin pumps often present problems related to weight gain since patients do not need to follow such a strict diet to keep BG levels within limits [84]. Turrin and Trujillo observed that one in five insulin-dependent diabetics using an insulin pump could not calculate the CHO content based on the nutritional label, and the same proportion of patients could not calculate the bolus insulin based on CHO intake and glucose values [84]. These findings point out that some patients rely entirely on smart pumps for calculating insulin doses, which is worrisome.

Christensen and colleagues [85] conducted a clinical study with 79 participants over 24 weeks to verify if using an ABC led to a better HbA1c when compared with the manual method of bolus calculation. The participants were T2DM patients treated with basal–bolus insulin therapy, all receiving training in CC for the first time. As a result, they obtained very similar improvements in HbA1c values for both cases, which indicates that CC is an excellent tool for managing the basal–bolus regimen in these patients, regardless of the method used to calculate the bolus.

Both insulin pumps with bolus wizard and ABC are great tools for helping diabetics; however, their efficacy is largely dependent on the patient’s ability to accurately estimate their CHO intake [86]. While discussing the potential benefits of advanced technologies in diabetes management, it is crucial to acknowledge the associated costs that may act as barriers to widespread adoption. According to the study of Addala et al., the uptake of diabetes technology has shown an increasing trend in the last decade [87]; however, population and registry data reveal a differential uptake, particularly among youth from lower socio-economic status families. This disparity is closely linked to the most commonly cited barrier to diabetes technology use—cost. Financial considerations play a significant role in shaping the landscape of technology adoption.

In the last decade, several apps emerged to help DM patients to estimate CHOs. However, there are still only a few patients who use this technology. This study addressed some examples of apps intended to automatically detect the CHO content of a meal. Despite presenting good accuracy, these applications have some limitations that are important to highlight. Food recognition is linked to the food database on which they are based. If a particular food is not part of this database, apps cannot respond. Another difficulty is related to the mixture of components on the plate. If the elements are not well separated or if the meal is a combination of foods, such as a pie or a lasagna, they cannot estimate correctly. All these point to the fact that it is essential to train the patient in CC.

Concerning education and training, there is no standard for learning goals or assessment. Currently, many diabetologists customize nutritional plans according to the individual’s dietary preferences and habits. However, training on CC is typically standardized. One limitation to this process is the tendency of most professionals to standardize the dietary plan and the literacy for CC. This generalization is considered an obstacle to the successful completion of this task. It is necessary to tailor the limits of the CC error and the learning objectives to maximize patient confidence and provide dietary freedom daily.

Also, accuracy assessment is a crucial part of the CC education process. Patients must be educated and trained until they are capable of accurately estimating the CHO content in a meal in order to maintain BG levels within safe limits. An inherent difficulty in evaluating patients’ errors in CC is the inability to know the exact amount of CHOs present in a meal. To overcome this issue, studies have considered the CHO value defined by a team of experienced nutritionists as the true value. Thus, the assessment of CC accuracy is usually performed in a controlled environment, but understanding its accuracy in a real-life context is a great challenge. Although the glycemic profile obtained from the CGM data helps to realize whether the CC was performed correctly, it is complex to isolate a single factor since many other factors can influence BG levels [88].

The studies carried out to assess the patients’ skills in this nutritional task confirmed the lack of standards, establishing different criteria and always generalized to the population studied. For example, in Meade and Rushton’s study [77], they considered that the patient correctly estimated the CHO content when the responses in the carbohydrate test were within a range of ±3 or 5 g depending on the total amount of carbohydrates contained in each item. Conversely, Gurnani et al. [78] established the limit of 10 g above or below the correct value of CHOs to be considered accurate. Deeb et al. determined that accuracy was within 20% of the estimation performed by experienced dietitians [79]. Therefore, assessment must be adjusted to the patient-specific requirements in the same way as the learning goals should be personalized.

### 5.2. Future Directions

It has become clear that to improve patients’ results in CC, training and assessment must be personalized. Intelligent algorithms and machine learning in the medical field are increasingly enhancing personalized medicine. As one of the most serious chronic illnesses in the world, diabetes is the subject of many scientific works in this regard. For example, combining AI with CGM and CSII devices allowed the development of the artificial pancreas (AP). The AP is a system that attempts to replicate the behavior of the pancreas in regulating BG levels. Even for this advanced system, it remains hard to control the glucose spike caused by the ingestion of CHOs.

Samadi et al. [89] proposed a solution to eliminate the need to manually insert meal announcements. Their system applies wavelet filtering, qualitative representation analysis, and fuzzy logic to the data from the CGM to detect meals and estimate their CHO content. In this regard, it is possible to determine the prandial insulin dosage according to the CHO intake without depending on the patient. However, the delay in the automatic meal detection leads to worse glycemic control than when patients announce meals.

Roversi and colleagues [88] studied the impact of different error levels in CC on the glycemic control of T1DM patients. To overcome the difficulty of isolating the effect of a single factor, they used the FDA-approved UVA/Padova simulator and developed in silico trials, varying the error in CC and keeping the other factors unchanged. The authors considered that with the mathematical formulas inferred through this study, the medical team could try to predict the improvement in glycemic control if the patient trained for a specific maximum error in CC. On the other hand, if a patient systematically makes a certain error, the physician can also adjust the patient’s ICR to compensate for the error.

In addition, it would be of great interest to assess the patient’s behavior in their daily eating routine. CGMs allow the monitoring of glycemia continuously, and these data have the potential to provide knowledge about the glycemic profile of an individual. It would be interesting to apply corrections to the general maximum error in CC so the patient could keep BG levels within limits considered safe for them. The use of machine learning techniques could be a plausible way to achieve this. However, using new technologies and AI to improve healthcare has an intrinsic concern, the fact that it is not possible to carry out tests that could harm the patient in any way. The UVA/Padova simulator is excellent for overcoming this difficulty in the case of type 1 diabetes, yet, it is limited when the goal is to customize the treatments even more.

Also, it might be interesting to apply the concept of a digital twin to this subject. Although the term digital twin is not new, this concept has only recently begun to be applied to the medical field. A digital twin refers to a digital copy or replica of a physical object or process. The existence of a virtual model that replicates specific characteristics of a patient has enormous potential for personalized medicine [90]. Ideally, a patient’s digital twin should include all relevant variables to pathogenesis, symptoms, and environmental factors [91]. This technology is already applied to diabetes, and it has demonstratedsuccessful results. Shamanna et al. [91] used digital twin technology, performing a retrospective study of T2DM patients enrolled in the twin precision nutrition (TPN) program. They aimed to compare HbA1c values and other glycemic metrics relevant to diabetes progression at baseline and 90 days later. The TPN program consists of AI and Internet of Things to create a virtual representation of a patient. This is possible with the use of body sensors and a mobile application that allows knowing how the metabolism of a given individual behaves, leading to the customization of the treatment. Through machine learning algorithms, the user received daily nutritional recommendations to avoid BG spikes. The authors concluded that patients who participated in this program had improvements in glycemic control, and for most of them, it was possible to stop taking diabetes medication.

## 6. Conclusions

Diabetes mellitus is a chronic disease with a high prevalence in the population, with a tendency to increase. Metabolic control is essential to prevent complications, manage the disease, and improve the quality of life for patients with DM. The treatment approach for this condition is based on three pillars, and finding a balance between them allows for optimal disease control. Regarding nutrition, the objective is to promote health and nutritional literacy, with the combination of advanced carbohydrate counting and insulin bolus administration being the most suitable approach to improve metabolic control. However, patient adherence to these strategies is not always ideal, and individualizing dietary plans is not yet a common practice.

The thorough exploration of CC education and assessment for diabetes management underscores its crucial role in achieving optimal glycemic control and improving patients’ quality of life. This study’s comprehensive analysis highlights the challenges that patients encounter in accurately applying this method despite recognizing the good outcomes of CC implementation. The heterogeneity of results emphasizes the need for personalized education and assessment, considering each patient’s unique circumstances and cultural background.

Technological advancements, such as insulin pumps and AI-powered applications, offer innovative tools to support patients in estimating CHO and administering insulin effectively. However, it is essential to strike a balance between leveraging technology and ensuring patients maintain a proactive role in managing their condition to prevent unintended consequences like weight gain. Also, these technologies may present some barriers such as device usability, the learning curve for patients adapting to new technologies, especially the older ones, and the impact on daily life routines.

Looking forward, integrating intelligent algorithms, machine learning, and digital twin technology holds great promise for refining the precision of carbohydrate estimation and insulin dosing. Yet, the adoption of intelligent algorithms and machine learning for refining carbohydrate estimation and insulin dosing introduces ethical considerations that should be carefully navigated in clinical decision making. This evolution in diabetes management signifies a shift towards personalized medicine, where patients receive customized treatment plans that consider their physiological responses, lifestyle factors, and behavioral patterns. Establishing personalized safety limits for CC errors would provide patients with a foundational basis for making informed nutritional decisions. This would enable them to estimate carbohydrate intake with increased confidence, as they would be aware of their individual boundaries. Additionally, the potential introduction of these personalized safe limits would empower healthcare practitioners to offer patients a more secure and precise framework for managing carbohydrates.

## Figures and Tables

**Table 1 nutrients-16-02183-t001:** Summary of diabetes management tools.

Tool	Technology	Main Purpose	Patient Intervention
Insulin pump bolus wizards	Calculator	Calculate bolus insulin	Input CHO estimation and BG values
Automatic insulin dose calculators	Calculator	Calculate bolus insulin	Input CHO estimation and BG values
GoCARBS [65]	Mobile application—based on computer vision	Estimate CHO content in a meal	Take pictures of the meal’s plate
BE(AR) [62]	Mobile application—based on augmented reality	Estimate CHO content in a meal	Point camera to meal’s plate and drawing
iSpy [63]	Mobile application—based on computer vision and voice recognition	Estimate CHO content in a meal	Take pictures of the meal’s plate or orally describe it
Intelligent Diabetes Management [66]	Mobile application	Diabetes diary and calculate bolus insulin	Input all the diabetes-related data (BG, CHO, exercise)
Glucose Buddy [67]	Mobile application—offering telemedicine	Connection with smart BG meter, generate reports and professional support anytime	Input meal nutritional information, body weight, and exercise
Diabetes Manager [64]	Mobile application	Calculate bolus insulin and record food CHO data	Input CHO estimation and BG values
Diabetes Diary [64]	Mobile application	Diabetes diary, food and medication advisor	Input CHO estimation and BG values
Dbees [68]	Mobile application	Diabetes diary, send reports to the doctor, set alarms	Input all the diabetes-related data (BG, CHO, exercise)
Diabetes Interactive Diary [69]	Mobile application—offering telemedicine	Carbohydrate/insulin bolus calculator, communication with health professionals	Select food picture and amount, input BG values
D-Partner [70]	Mobile application—offering telemedicine	Calculate bolus insulin, communication with health professionals, automatic notifications	Input all the diabetes-related data (BG, CHO, exercise)
VoiceDiab [71]	Mobile application—with automatic speech recognition	Calculate bolus insulin, estimate CHO content in a meal	Voice description of the meal

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
