# Peer review of "Assessing Carbohydrate Counting Accuracy: Current Limitations and Future Directions"

_nutrients, 2024, doi:10.3390/nu16142183_

Round 1
Reviewer 1 Report
Comments and Suggestions for Authors
Overall evaluation:
This review provides insights into the benefits and limitations of existing tools and identifies future research directions to advance personalized CC training approaches. By adopting a personalized approach to CC education and assessment, healthcare professionals can empower patients to achieve better glycemic control and improve diabetes management. The topic selection of the review is novel and the content of the review is comprehensive, which reflects the latest research progress and opinions of various people, and has a certain guiding role for related research.
Modification Suggestion:
1.Line23-23, the number of diabetic patients introduced in the manuscript is 2021, it is best to query the data replacement until 2023.
2. For Line29-32, data ranges of target Blood Glucose (BG) levels and normal should be specified. In addition, why people with diabetes experience Hypoglycemia needs to be explained.
3.Line106-108, CHO is equal to digestible carbohydrate, please add clarification.
4.Line116-127, it is recommended to elaborate the calculation process of Basic Carbohydrate Counting and Advanced Carbohydrate Counting, as well as the difference between the two methods.
The headings of 5.2.1 and 2.2 are too simple and should be appropriately supplemented to make the meaning of the headings specific.
6. The main content of this paper is about Carbohydrate Counting and T1DM, which is mainly T2DM in diabetes mellitus. It is suggested to add discussion on the relationship between Carbohydrate Counting and T2DM.
7. The content of the article is more detailed, but all are text summaries. In order to improve the absorption of the article, it is suggested that the author add graphs or tables.
8. Glycemic index (GI) and glycemic load are closely related to human blood sugar health, and it is recommended to properly supplement Carbohydrate Counting and their correlation.
Author Response
Comments 1: Line23-23, the number of diabetic patients introduced in the manuscript is 2021, it is best to query the data replacement until 2023.
Response 1: Thank you for pointing this out. We have been searching for more recent statistics, but we couldn’t find official data after 2021. So, we decided to maintain the text as it is.
Comments 2: For Line29-32, data ranges of target Blood Glucose (BG) levels and normal should be specified. In addition, why people with diabetes experience Hypoglycemia needs to be explained.
Response 2: We agree with this comment. Therefore, we have made the changes on page 2, lines 32, 36-39.
Comments 3: Line106-108, CHO is equal to digestible carbohydrate, please add clarification.
Response 3: Thank you for pointing this out. We added the clarification on page 2, line 50.
Comments 4: Line116-127, it is recommended to elaborate the calculation process of Basic Carbohydrate Counting and Advanced Carbohydrate Counting, as well as the difference between the two methods.
Response 4: thank you for pointing this out. We added the formula to calculate bolus insulin in Advanced Carbohydrate Counting (page 4, line 145-148). As the basic system doesn’t include bolus insulin, we couldn’t include anything new.
Comments 5: The headings of 5.2.1 and 2.2 are too simple and should be appropriately supplemented to make the meaning of the headings specific.
Response 5: We agree with this comment. Therefore, we made changes to those titles to be more comprehensive (lines 155 and 174).
Comments 6: The main content of this paper is about Carbohydrate Counting and T1DM, which is mainly T2DM in diabetes mellitus. It is suggested to add discussion on the relationship between Carbohydrate Counting and T2DM.
Response 6: Thank you for your suggestion. This review focuses on cases requiring insulin therapy, which are primarily found in T1DM. Therefore, the literature consulted and the information included in this review are centered on Type 1 Diabetes.
Comments 7: The content of the article is more detailed, but all are text summaries. In order to improve the absorption of the article, it is suggested that the author add graphs or tables.
Response 7: We agree, and we have, accordingly, added a table on page 10.
Comments 8: Glycemic index (GI) and glycemic load are closely related to human blood sugar health, and it is recommended to properly supplement Carbohydrate Counting and their correlation.
Response 8: Thank you for pointing this out. We acknowledge the importance of these concepts; however, a detailed discussion on this topic falls outside the scope of this review.
Reviewer 2 Report
Comments and Suggestions for Authors
This outstanding review investigates the carbohydrate counting technique. The article is comprehensive and well-written, offering solutions to current problems.
- Section 3-4: I would propose to add a table that summarizes the advantages and disadvantages of the most important tools that are described.
- Section 3: not all apps are validated, which should be noted by the authors.
- Section 5: more recently, counting of the actual grams (instead of carbohydrate portions) is making way, as this allows for a better glycemic control, which should be discussed.
Comments on the Quality of English LanguageSome grammatical errors throughout the text.
Author Response
Comments 1: [Section 3-4] I would propose to add a table that summarizes the advantages and disadvantages of the most important tools that are described.
Response 1: Thank you for pointing this out. We agreed, so we added a summary table on page 10.
Comments 2: [Section 3] not all apps are validated, which should be noted by the authors.
Response 2: We agree, so we have, accordingly, added that information in lines 376-378.
Comments 3: [Section 5] more recently, counting of the actual grams (instead of carbohydrate portions) is making way, as this allows for a better glycemic control, which should be discussed.
Response 3: Thank you for your valuable input. We did not include this information in the discussion because none of the studies reviewed provided evidence supporting this claim.